# Development and Validation of a Method for Quantification of Favipiravir as COVID-19 Management in Spiked Human Plasma

**DOI:** 10.3390/molecules26133789

**Published:** 2021-06-22

**Authors:** Mohammad Hailat, Israa Al-Ani, Mohammed Hamad, Zainab Zakareia, Wael Abu Dayyih

**Affiliations:** 1Faculty of Pharmacy, Al-Zaytoonah University of Jordan, Amman 11733, Jordan; m.hailat@zuj.edu.jo; 2Faculty of Pharmacy, Al-Ahliyya Amman University, Amman 19328, Jordan; issrahamid@gmail.com; 3College of Science and Health Professions, King Saud Bin Abdulaziz University for Health Sciences, Jeddah 21423, Saudi Arabia; hamadm@ksau-hs.edu.sa; 4Faculty of Pharmacy and Medical Sciences, University of Petra, Amman 11196, Jordan; zzakareia@uop.edu.jo

**Keywords:** bio-analytical, validation, HPLC, spiked human plasma, acyclovir, favipiravir

## Abstract

In the current work, a simple, economical, accurate, and precise HPLC method with UV detection was developed to quantify Favipiravir (FVIR) in spiked human plasma using acyclovir (ACVR) as an internal standard in the COVID-19 pandemic time. Both FVIR and ACVR were well separated and resolved on the C18 column using the mobile phase blend of methanol:acetonitrile:20 mM phosphate buffer (pH 3.1) in an isocratic mode flow rate of 1 mL/min with a proportion of 30:10:60 %, *v/v/v*. The detector wavelength was set at 242 nm. Maximum recovery of FVIR and ACVR from plasma was obtained with dichloromethane (DCM) as extracting solvent. The calibration curve was found to be linear in the range of 3.1–60.0 µg/mL with regression coefficient (r^2^) = 0.9976. However, with acceptable r^2^, the calibration data’s heteroscedasticity was observed, which was further reduced using weighted linear regression with weighting factor 1/x. Finally, the method was validated concerning sensitivity, accuracy (Inter and Intraday’s % RE and RSD were 0.28, 0.65 and 1.00, 0.12 respectively), precision, recovery (89.99%, 89.09%, and 90.81% for LQC, MQC, and HQC, respectively), stability (% RSD for 30-day were 3.04 and 1.71 for LQC and HQC, respectively at −20 °C), and carry-over US-FDA guidance for Bioanalytical Method Validation for researchers in the COVID-19 pandemic crisis. Furthermore, there was no significant difference for selectivity when evaluated at LLOQ concentration of 3 µg/mL of FVIR and relative to the blank.

## 1. Introduction

The coronavirus disease 2019 (COVID-19) has resulted in hospitalizations for many people due to pneumonia with the multiorgan disease [1]. COVID-19 is the cause of the novel severe acute respiratory syndrome coronavirus 2 (SARS-CoV-2), ending in many deaths worldwide. The disease is transmitted mainly by respiratory droplets from person to person [1]. Favipiravir (FVIR, Avigan^®^) was among the pioneer medications used against SARS-CoV-2 in Wuhan at the pandemic’s very core. FVIR (T-705) is a synthetic prodrug found to assess the antiviral activity of hits that had activity against the influenza virus in the chemicals under investigation in Toyama Chemical. One of the chemicals under investigation showed promising results. This hit, A/PR/8/34, later designated as T-1105, and related derivatives were discovered to have antiviral activities. FVIR is one of the drugs derived from the pyrazine moiety of T-1105 (Figure 1) [2,3]. In 2014, it was approved in Japan to manage emerging pandemic influenza infections. Chemically, FVIR is (6-fluoro-3-hydroxypyrazine-2-carboxamide) an analog of pyrazine “C_5_H_4_FN_3_O_2_” (Figure 1a).

FVIR could be given as a prodrug. This prodrug has high bioavailability (~94%), 54% protein binding, and a low distribution volume of around 10–20 L. It reaches C_max_ within two h after first administration. T_max_ and t_1/2_ were observed to elevate after multiple doses. FVIR has a short t_1/2_ of about 2.5 to 5 h, resulting in rapid kidney elimination in the hydroxylated form. Elimination starts with aldehyde oxidase and slightly with xanthine oxidase. The pharmacokinetic profiles of FVIR are dose-dependent and time-dependent profiles. Simultaneously, it is not metabolized by the CYP P450 system; it inhibits one of its components (i.e., CYP2C8). Therefore, cautiously, it should be used when coadministered with drugs metabolized by the CYP2C8 system [4,5]. Acyclovir, C_8_H_11_N_5_O_3_, Figure 1b, is an acyclic purine nucleoside analog used as an internal standard for chromatographic analytical methods to validate variability in sample processing analysis [6,7].

HPLC is a frequently used method to analyze medications, either alone or as a combination [8,9,10]. Several HPLC methods were reported to estimate FVIR either in the dosage forms [2] or different biological fluids [11,12]. Further, an HPTLC method was reported to estimate FVIR in human plasma with acyclovir (ACVR) as an internal standard [13]. Along with these, several hyphenated techniques were reported for the estimation of FVIR alone [13,14,15,16], including an LC-MS method, where FVIR was extracted using liquid–liquid extraction and estimation of FVIR and other antiviral drugs [3,14].

Hence, considering the need for a simple, economically accurate, precise, and selective method for estimating FVIR in human plasma, an attempt was made to develop an HPLC method for estimating FVIR in human plasma [15]. FVIR was extracted from plasma by liquid–liquid extraction using a suitable organic solvent. All parameters for analysis were selected considering the C_max_ of the drug. The heteroscedasticity observed in the calibration data was reduced by weighted linear regression with a suitable weighting factor [15,16,17,18,19]. However, the simplicity of our analysis method, compared with other methods [15], lacks the need for expert people to carry out the analysis and does not need biomedical infrastructure. Our manuscript is new relative to other work using spiked human plasma and ACVR as an internal standard. Indeed, this method is a simple, accurate, and precise HPLC method applicable easily for Human Plasma. Furthermore, our method was validated based on US-FDA regulations for Bioanalytical Method Validation [19,20].

## 2. Results and Discussion

### 2.1. Optimized Chromatographic Condition

Different strengths of mobile phases were prepared to obtain adequate retention with acceptable system suitability. The mobile phase with the methanol composition: acetonitrile: 20.0 mM phosphate buffer (pH 3.1) in a proportion of 30:10:60 % *v*/*v*/*v* gave adequate separation and resolution when used in an isocratic mode at a flow rate of 1 mL/min. The separation was obtained on Symmetry^®^ C18-(250 cm × 4.6 mm, 5 μm an average particle size) (Waters Corp., Dublin, Ireland). All eluents were detected at 242 nm. The retention time for FVIR and ACVR was found at 7.40 min and 4.64 min, respectively. However, FAVIR in another study was analyzed using LC-2030 C system equipped with Shim-Pack GIST C18 (250 × 4.6 mm, 5 μm) column using a mobile phase mixture of 10 mM phosphate buffer (pH 4.0) and acetonitrile in the ratio of 90:10 *v*/*v* at a flow rate of 1.0 mL/min., and the run time was 8.0 min [21]. In another study that analyzed FAVIR using a C18 column, the mobile phase was a mixture of 50 mM phosphate buffer (pH 2.3) and acetonitrile (90:10, *v*/*v*) and the flow rate was 1 mL/min, and the run time was 15 min under these conditions [2].

### 2.2. Optimization of Liquid–Liquid Extraction (LLE) Experiment

Different organic solvents were investigated for the extraction of FVIR and ACVR from plasma. As depicted in Table 1, no extraction for FVIR and ACVR was observed in n-hexane, tetrahydrofuran, and toluene. However, suitable extraction of 91.31% and 93.69% was obtained in dichloromethane (DCM), in a well-ventilated room, for FVIR and ACVR, respectively. The representative chromatogram of blank plasma is presented in Figure 2, and the representative chromatogram of FVIR and ACVR extracted in DCM is presented in Figure 3. Further, 5 mL of organic solvent and 4000 rpm speed for centrifuge were found optimum.

### 2.3. Selection of Internal Standard

Out of the different analytes injected to select internal standard, ACVR gave adequate resolution from the plasma interferents and FVIR with acceptable system suitability. Hence, ACVR was selected as an internal standard in this study. Additionally, when different concentrations of ACVR were injected in HPLC with the highest concentration of FVIR, it was found that a concentration of 18.0 µg/mL gave an acceptable peak area with that of FVIR.

### 2.4. Calibration Curve Study and Selection of Regression Model

When the obtained CC data (Table 2) were subjected to unweighted linear regression, an acceptable r^2^ of 0.9974 was observed with the CC equation of y = 3.38 × 10^−5^x + 0.0026. However, when the CC data were subjected to a homoscedasticity test, the F calculated was more significant than F theoretical. This suggests the necessity for weighted linear regression. Hence, the CC data were subjected to weighted linear regression with different weighting factors, 1/x, 1/x^2^, 1/√x, 1/y, 1/y^2^, and 1/√y. The regression analysis of FVIR with different weighting factors is shown in Table 3. Thus, it was found that weighted linear regression with weighting factor 1/x showed minimum % RE and was further used for calculations.

### 2.5. Method Validation

When the selectivity was evaluated at the LLOQ concentration of 3 µg/mL of FVIR and compared with the peak areas of blank plasma at the retention time of FVIR, no significant peaks in the chromatograms of blank plasma were observed at the retention time of FVIR. Thus, the response of blank plasma samples was found less than 20% to the LLOQ. The response of each plasma sample with that of FVIR is depicted in Table 4.

The results of accuracy, precision, minimum % RE and % RSD studies are shown in Table 5. At each QC level, i.e., LQC, MQC, and HQC levels, it was proved that the accuracy and precision of the method within the selected CC range.

The recovery data presented in Table 6 proved the acceptable recovery of FVIR and ACVR within the given experimental conditions.

The results of stability studies are presented in Table 7, Table 8 and Table 9. From these data, it can be concluded that the % nominal values were between 85–115%, and the % RSD values were less than 15% for all the stability samples, which is acceptable according to EMEA guidelines for bioanalytical method validation [22]. This proved that the drug remained stable after the completion of the stability cycles.

A carry-over study was conducted as per the mentioned sequence (Table 10). However, no residue of FVIR with previous samples was observed in the subsequent runs. Thus, it can be concluded that no carry-over effect was seen in the developed method.

## 3. Materials and Methods

### 3.1. Chemicals and Reagents

Pharmaceutical grade FVIR (Favipiravir, USP) was purchased from the local Market. Sancovir^®^ 200mg/tablet recently obtained approval by Jordan Food and Drug Administration JFDA, supplied by Al Wafi group, Amman, Jordan, manufactured by Sana Pharma Amman, Jordan and certified to contain 99.6% *w*/*w* an anhydrous basis. Blank human plasma from different sources was obtained as a gift sample from Royal Hospital Amman, Jordan, and the pooled sample was prepared by vigorously mixing the obtained samples of plasma. The acetonitrile and methanol were of HPLC grade, and the rest of the chemicals used were of analytical reagent grade. All chemicals were purchased from Merck Life Sciences Pvt. Ltd., Darmstadt, Germany. In addition, freshly prepared double distilled water used in the analysis was obtained using all Glass Distillation Assembly, purchased from Kilo Lab—Über W. Köpp GmbH and Co. KG Millipore—Merck Millipore Billerica, MA, USA.

### 3.2. Instrumentation and Chromatographic Conditions

The HPLC system used was Hitachi Chromaster system (Hitachi High-Tech science Corp., Tokyo, Japan). This HPLC system is equipped with a 5410 UV detector, 5260 autosamplers, 5310 Column oven, and 5160 quaternary pumps. The column used to achieve the separation was Symmetry^®^ C18-(250 cm × 4.6 mm, 5 μm an average particle size) (Waters Corp., Dublin, Ireland). The chromatographic data analysis was performed using Clarity Chromatography Station (Chromatography Station for Windows, version 8.1, DataApex, Podohradska, Czech Republic). The weighing was performed on AUX 220 digital weighing balance, Shimadzu Corporation, Tokyo, Japan. C-24 BL, cooling centrifuge used in analysis 8KBS Three Phase Air Cooled Centrifuge

A compact refrigerated floor standing centrifuge for universal use in blood banks and clinical laboratories was purchased from Sigma Laborzentrifugen™, Germany. FVIR and internal standard (IS), ACVR were separated and resolved from each other, and the plasma interferes using a blend of methanol: acetonitrile: 20.0 mM phosphate buffer (pH 3.1) (30:10:60 %, *v*/*v*/*v*) as a mobile phase in an isocratic mode with a flow rate of 1.0 mL/min. All eluents were detected at 242 nm, the absorbance maxima of FVIR.

### 3.3. Preparation of Standard Stock Solutions

The standard stock solution of 1 mg/mL of FVIR and ACVR was obtained by dissolving 100 mg FVIR and ACVR individually in a 100 mL volumetric flask using methanol. The prepared standard stock solution of FVIR was further diluted with methanol to obtain ten different working standard solutions of concentrations 30, 60, 90, 150, 250, 300, 400, 450, 550, and 600 µg/mL. Additionally, the standard stock solution of ACVR is diluted accordingly with methanol to obtain a concentration of 180 µg/mL.

### 3.4. LLE Experiment

An aliquot of 1 mL of pooled plasma was taken in a glass tube with a stopper of 20.0 mL size in the LLE experiment. In it, 100 µL of 100 µg/mL of FVIR and 100 µL of 100 µg/mL of ACVR (IS) were added, and the solution was vortex mixed for 5 min. Further, an aliquot of 5.0 mL of DCM was added to it, and the sample in the tube was vortex remixed for 5 min. The tube was then centrifuged at 4000 rpm for 10 min at 5.0 °C in a cooling centrifuge. Next, the separated organic layer was added to an Eppendorf tube and evaporated to dryness under nitrogen. The residue obtained after this was then reconstituted with 500 µL of the mobile phase. Finally, an aliquot of 20 µL of this solution was injected into the chromatographic system.

### 3.5. Preparation of Calibration Curve (CC) Standard and Quality Control (QC) Samples

We were considering the C_max_ of the FVIR (25–45 µg/mL) [23] where the maximum plasma concentration occurred at two hours after oral administration; 30 µg/mL was taken for the experiment; the CC standards and QC samples were prepared according to the US-FDA guidelines for Bioanalytical Method Validation. Hence, the CC standards were prepared in the range of 3–60 µg/mL. Additionally, QC samples, which includes LLOQ—3 µg/mL (10% of C_max_), LQC—9 µg/mL (3 times the LLOQ), MQC—30 µg/mL (30—50% of the calibration range), HQC—45 µg/mL (near to the upper limit of CC range) were prepared.

The CC standards were prepared by taking 1 mL of an aliquot of pooled plasma in 10 different stoppered glass tubes of size 20 mL. Individually in each tube, 100 µL of prepared working standard solutions of FVIR and 100 µL of 180 µg/mL of ACVR was added to obtain CC standards of 3, 6, 9, 15, 25, 30, 40, 45, 55, and 60 µg/mL of FVIR, respectively.

All calibration curves and standard solutions were processed as per the procedure depicted in the LLE experiment section and finally injected into the HPLC system under mentioned chromatographic conditions.

The QC samples of concentrations 9 µg/mL (LQC), 30 µg/mL (MQC), and 45 µg/mL (HQC) were prepared along with CC standard similarly.

### 3.6. Selection of Internal Standard

Different analytes with similar chromatographic behavior to FVIR were observed in this research, and the ACVR showed the optimum as an internal standard. The analyte that showed good resolution from the FVIR and plasma interferences and acceptable system suitability was selected as an internal standard. Further, to select the concentration of the IS, different concentrations of selected IS were injected in the HPLC system with the highest concentration of FVIR (i.e., 60 µg/mL) and the IS concentration, which gave 30–60% peak area to that of the highest concentration of FVIR selected.

### 3.7. Calibration Curve and Selection of Calibration Model

All points in the CC standard were injected in six replicates. The obtained chromatograms of all CC standards were integrated, and the peak area ratio for FVIR to ACVR was calculated. The obtained peak area ratio for each CC standard was plotted against the respective concentration to construct a calibration curve.

Further, the obtained data from the CC standards were subjected to unweighted and weighted linear regression. Different weighting factors, 1/x, 1/x2, 1/√x, 1/y, 1/y2, and 1/√y, were evaluated, and the calibration model with minimum % relative error (% RE) and uniform scatter of points in the residual plot were selected and used in further calculations.

### 3.8. Method Validation

The developed method was validated according to the US-FDA guidelines for Bioanalytical Method Validation [24,25].

Selectivity was evaluated at a lower limit of quantitation (LLOQ) at a concentration of 3 µg/mL (10% of C_max_), where the sample of LLOQ was analyzed, peak area was noted and compared with the response obtained for the blank plasma sample at the retention time of FVIR. The experiment was performed six times for each source of a plasma sample. The method’s accuracy and precision were accessed by recording the % RE and % RSD, respectively, for five replicates of LQC, MQC, and HQC samples for five successive days. The recovery study was performed by comparing the processed QC samples’ peak areas with the standard dilutions representing 100% recovery in five replicates. The samples’ stability was studied at the ambient temperature, at −20 °C, benchtop stability, freeze–thaw stability, and long-term stability. For each type of stability study, the % nominal and % RSD values were calculated. To evaluate the carry-over between samples, a series of samples were injected into the HPLC system, and the residue of the previous samples was observed in the subsequent sample.

## 4. Conclusions


A simple, rapid, accurate, precise, and selective HPLC method was described to quantify FVIR in human plasma in the present work.Liquid–liquid extraction provided an excellent recovery with clear FVIR, and ACVR (internal standard) extract from plasma using DCM.FVIR and ACVR were well separated and resolved from each other and plasma interferents on the C18 column using methanol:acetonitrile:20.0 mM phosphate buffer (pH 3.1) (30:10:60 % *v*/*v*/*v*) in an isocratic mode at a flow rate of 1 mL/min. The method proved to be economical as the total run time per sample was less than 10 min. All eluents were detected at 242 nm.When the calibration data were subjected to linear regression, despite an acceptable r^2^ of 0.9974, the calibration data were susceptible to heteroscedasticity, leading to an error at a higher concentration level.To reduce the heteroscedasticity, weighted linear regression models were implemented with different weighting factors, and the weighing factor of 1/x proved to give acceptable results with minimal % RE.The developed method was validated according to the US-FDA guidelines for Bioanalytical Method Validation. In May 2018, acceptable selectivity, accuracy, precision, recovery, sample stability, and carry-over were obtained within the studied calibration range.


## Figures and Tables

**Figure 1 molecules-26-03789-f001:**
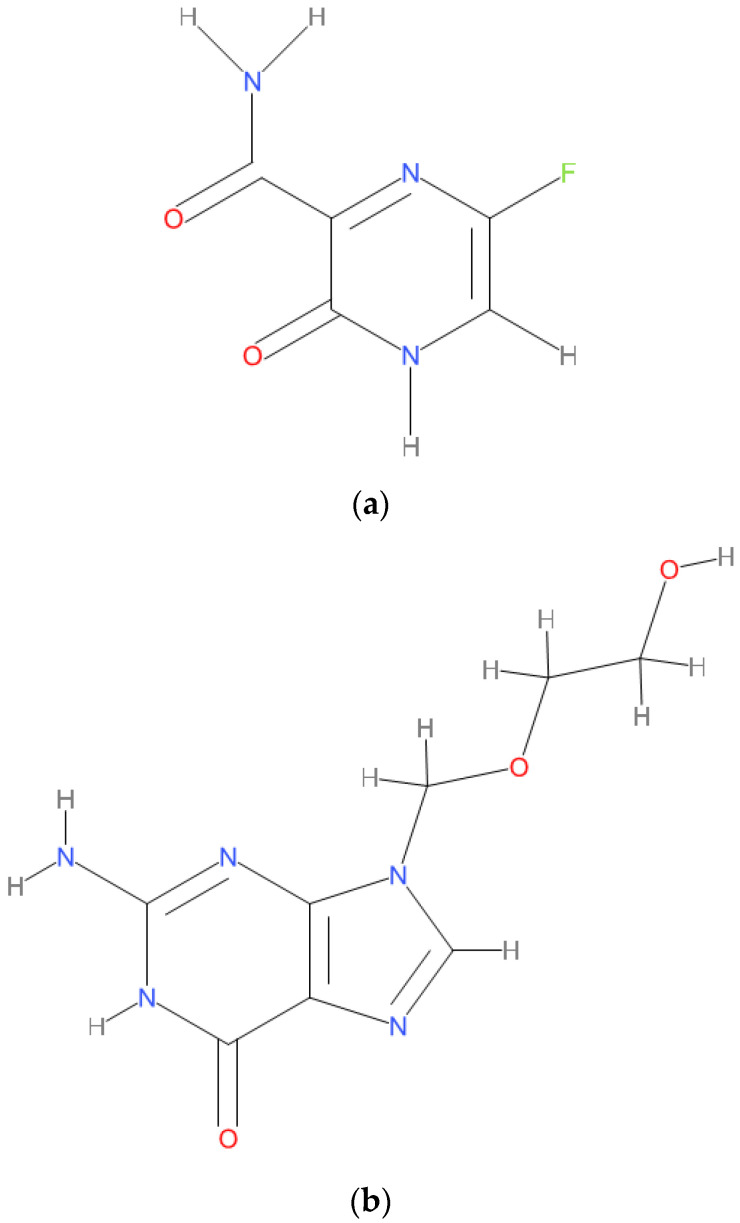
(**a**) Chemical Structure of Favipiravir. (**b**) Chemical Structure of Acyclovir.

**Figure 2 molecules-26-03789-f002:**
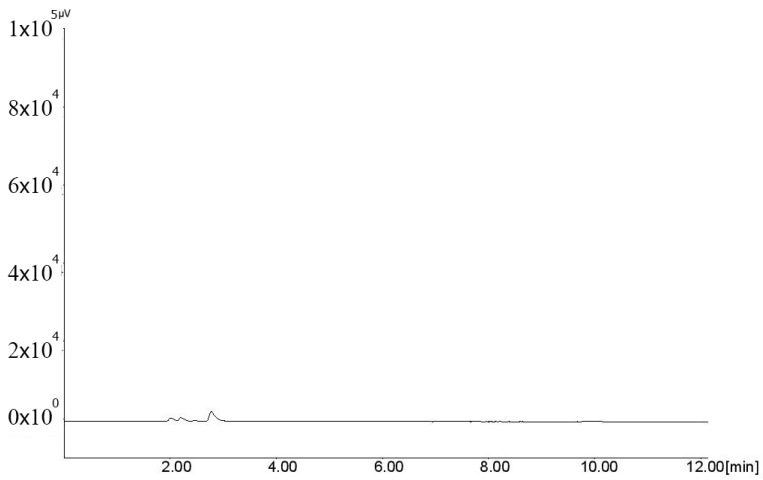
Representative chromatogram of blank plasma extracted with dichloromethane.

**Figure 3 molecules-26-03789-f003:**
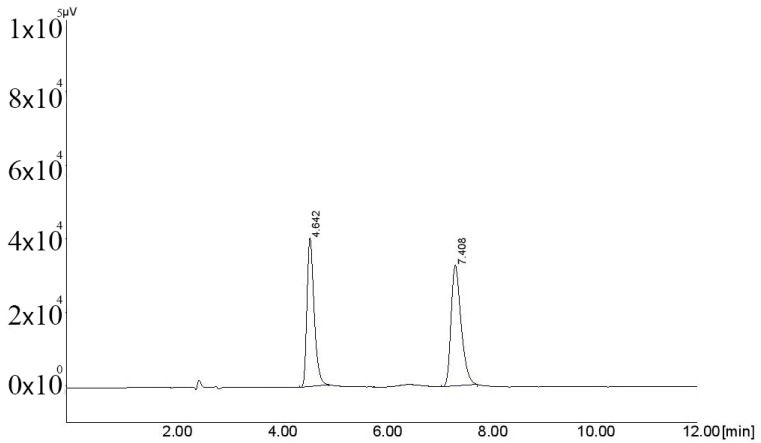
Representative chromatogram of FVIR and ACVR (IS) extracted in dichloromethane. (FVIR RT: 7.40 min; IS RT: 4.64 min).

**Table 1 molecules-26-03789-t001:** The recovery of FVIR and IS in different organic solvents.

Sr. No.	Organic Solvent Used for LLE.	% Recovery of FVIR	% Recovery of ACVR
1	n-Hexane	-	-
2	Tetrahydrofuran	-	-
3	Toluene	-	-
4	Chloroform	38.43	59.34
5	Chloroform + 1% Formic acid	42.43	63.41
6	TBME (tert-butyl methyl ether)	52.34	53.87
7	Ethyl Acetate	55.25	56.18
8	TBME + 5%Formic acid	71.49	78.33
9	TBME + 1%Formic acid	72.25	78.12
10	TBME + Ethyl acetate + 1%Formic acid	80.51	84.06
11	DCM	**91.31**	**93.69**

**Table 2 molecules-26-03789-t002:** Calibration curve (CC) data for FVIR.

Sr. No.	Concentration (µg/mL)	Area Ratio (Mean ± SD) *n* = 6	% RSD
1	3	0.12 ± 0.0026	2.12
2	6	0.21 ± 0.0044	2.04
3	9	0.32 ± 0.0049	1.51
4	15	0.46 ± 0.0035	0.76
5	25	0.82 ± 0.0037	0.46
6	30	1.05 ± 0.0026	0.25
7	40	1.32 ± 0.0038	0.29
8	45	1.58 ± 0.0024	0.15
9	55	1.89 ± 0.0052	0.27
10	60	1.99 ± 1.0061	0.31

**Table 3 molecules-26-03789-t003:** Weighted linear regression of FVIR with different weighting factors.

Sr. No.	Weighing Factor	Intercept (a)	Slope (b)	r^2^	% RE
1	1	0.004900433	0.00003381	0.998	88.70
2	1/x	0.010899824	0.00003345	0.999	64.05
3	1/x2	0.017234891	0.00003288	0.999	73.72
4	1/√x	0.007124143	0.00003370	0.999	83.36
5	1/y	0.009318948	0.00003345	0.999	88.43
6	1/y2	0.015485202	0.00003279	0.999	116.92
7	1/√y	0.006283860	0.00003366	0.998	94.67

**Table 4 molecules-26-03789-t004:** Blank response and peak areas of FVIR at LLOQ.

Sr. No.	Blank Response (µV·s)	Peak Areas at LLOQ (µV·s)	% Peak Area in the Blank
1	4972.16	441,119.25	1.12%
2	5323.44	437,215.64	1.21%
3	5194.26	445,018.86	1.16%
4	6437.38	429,986.93	1.49%
5	6198.91	450,128.16	1.37%
6	7011.21	435,625.45	1.60%

**Table 5 molecules-26-03789-t005:** Results of assessment of the accuracy and precision studies of FVIR.

QC Level	Conc. Added (µg/mL)	Inter day (*n* = 5) Mean Conc. Found (µg/mL)	% RE	% RSD	Intraday (*n* = 5) Mean Conc. Found (µg/mL)	% RE	% RSD
LQC	9	9.03	0.28	0.65	9.09	1.00	0.21
MQC	30	30.58	1.94	1.66	31.01	3.38	0.02
HQC	45	44.80	−0.45	5.33	46.07	−1.19	0.02

**Table 6 molecules-26-03789-t006:** Recovery of FVIR and ACVR.

Samples	Peak Area of FVIR			Peak Area of ACVR (μV·s)
	QC Levels			
	LQC	MQC	HQC	
**Unextracted**	169,253.41	582,099.95	782,553.28	566,705.68
**Extracted**	152,310.49	518,536.08	710,582.72	520,469.25
**% Recovery**	89.99%	89.09%	90.81%	91.84%

**Table 7 molecules-26-03789-t007:** The stability of FVIR at room temperature and −20 °C.

QC Level	Stability at Room Temperature						Stability at −20 °C					
	% Nominal			% RSD			% Nominal			% RSD		
	2 h	4 h	6 h	2 h	4 h	6 h	10 days	20 days	30 days	10 days	20 days	30 days
LQC	98.8	98.4	98.7	1.91	1.17	1.51	99.7	99.1	99.7	1.01	1.12	3.04
HQC	101	100	101	2.33	1.49	1.60	100	101	101	1.50	1.27	1.71

**Table 8 molecules-26-03789-t008:** Freeze–thaw stability of FVIR.

	Freeze–Thaw Stability					
	% Nominal			% RSD		
QC Level	FT1	FT2	FT3	FT1	FT2	FT3
LQC	99.83	99.35	99.45	0.16	0.28	0.33
HQC	101.28	98.23	100.98	0.04	0.04	0.04

**Table 9 molecules-26-03789-t009:** The benchtop and long-term stability study results.

QC Level	Benchtop		Long Term	
	%Nominal	% RSD.	%Nominal	% RSD
	8 h	8 h	30 days	30 days
LQC	99.80	0.10	100.42	0.07
HQC	100.04	0.09	101.43	0.02

**Table 10 molecules-26-03789-t010:** Results for carry-over study.

Sr. No.	Sample	Area (μV·s)	
		FVIR	ACVR
1	Blank solution	0	0
2	Unextracted ULOQ	467,486.79	559,303.58
3	Blank solution	0	0
4	Unextracted ULOQ	459,387.84	568,700.87
5	Blank solution	0	0
6	Extracted blank plasma	0	0
7	Extracted ULOQ	416,063.25	505,964.47
8	Extracted blank plasma	0	0
9	Extracted ULOQ	409,893.75	512,348.50
10	Extracted blank plasma	0	0

## Data Availability

The data presented in this study are available on request from the corresponding author.

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
