# Peer review of "Development and Validation of a Method for Quantification of Favipiravir as COVID-19 Management in Spiked Human Plasma"

_molecules, 2021, doi:10.3390/molecules26133789_

Round 1

Reviewer 1 Report

The authors have adequately address all of the points raised by the Reviewer. However, I found the following minor points in the revised manuscript. 

1) Line 33, "several people": Why is it only "several"?

2) Line 88, "ACN": This should be "acetonitrile".

3) Line 89-90, "The further study analyzed FAVIR was using a C18 column": This should be "A further study analyzed FAVIR using a C18 column".

4) Line 92, "1 mL/min-1": "-1" should be deleted. 

Author Response

Response to Reviewer 1 Comments

Great thanks for taking the time to review and improve our manuscript. Your valuable comments are highly appreciated. Reviewer comments, suggestions, and feedback helped us improve the manuscript substantially.

Point 1: Line 33, "several people": Why is it only "several"?

Response 1: We appreciate the reviewer’s insightful suggestion and agree with it. The phrase "several people" has changed to “many people”. Thank you.

Point 2: Line 88, "ACN": This should be "acetonitrile".

Response 2: This observation is correct. The word “ACN” has changed accordingly to “acetonitrile”. Thank you for this comment.

Point 3: Line 89-90, "The further study analyzed FAVIR was using a C18 column": This should be "A further study analyzed FAVIR using a C18 column".

Response 3: We thank the reviewer for pointing this out. We have changed the phrase from "The further study analyzed FAVIR was using a C18 column" to "A further study analyzed FAVIR using a C18 column". Thank you.

Point 4: Line 92, "1 mL/min-1": "-1" should be deleted.

Response 4: We agree with the reviewer's comment and apologize for the error. The "-1" has been deleted accordingly.

Finally, we would like to conduct our recognition and respect for the reviewer's effort and time to improve the manuscript and clean it—many thanks.

Reviewer 2 Report

Still, it is not mentioned what is really new in comparison with similar papers from the literature. Thus, the scientific value of the present paper is unclear. I do not find this answer exhaustive:

„Our manuscript is new relative to other work using spiked human plasma and using acyclovir (ACVR) as an internal standard. Indeed, this method is a simple, accurate, and precise HPLC method applicable easily for Human Plasma.”

What is more, the self-citations (Refs. 8-10) are still unclear. I am not fully convinced.

Two different columns were mentioned:

„The separation was obtained on Phenomenex Hyperclone C18 (250 × 4.6 mm, 45μ).”

„The column used to achieve the separation was Symmetry®C18-(250cm×4.6mm, 5μm an average particle size) (Waters Corp., Ireland).”

What does it mean:

„The chromatographic data analysis was performed using clarity VA chromatography”.

The Ref. 2 and Ref. 22 are the same.

Some sentences uare unclear, e.g.

„The further study analyzed FAVIR was using a C18 column,and the mobile phase was…”

„At each QC level proved the accuracy and precision of the method within the selected CC range.”

Reviewer 3 Report

The revised version of this manuscript shows significant amendment. Authors have presented well method deveopment and validation. Some minor comments still:

-In Table 3, the additional digit for R^2 should be removed, i.e. 0.9976 should be 0.998. We need the first digit that is not 9

-In Introduction authors should make more clear the superiority of their method

Author Response

This manuscript is a resubmission of an earlier submission. The following is a list of the peer review reports and author responses from that submission.

Round 1

Reviewer 1 Report

Favipiravir (trade name, Avigan) is considered to be effective against COVID-19 viruses. Therefore, it is desirable to develop a precise, rapid, and economical method to analyze its level in plasma. In the present study, the authors have developed such a method using solvent extraction followed by HPLC-UVD. The method appears to be applicable to clinical settings. However, there are a number of flaws in the manuscript, which need to be addressed before acceptance.

Major points:

  • Dichloromethane used for LLE is a toxic solvent and thus must be used with a good ventilation. This point should be mentioned.
  • The authors should pay more attention on “significant digits or figures” throughout the text. For example, 12.3456 should be 12.35 (or 12.3). This applies to most of the numerical data.
  • Abbreviations should be full-spelled at the first appearances. For example, LLE should be liquid-liquid extraction (LLE) in line 79.
  • References: In line 61 and 69, references 7-26 and 29-37 are cited without referring to each reference. References should be selected only when they are relevant to the present study and brief comments should be given.

Minor points:

  • Line 33: “Toyoma Chemicals” should be “Toyama Chemical”.
  • Line 82: Extraction of 89.72 and 90.21% was obtained. But in Table 2, they are 89.72 and 90.21. Why?
  • Line 98: “Hence, selected as an internal standard in this study.” This does not make a sentence.
  • Table 3: r2 values of 0.998 should be given one more digit such as 0.9983 because they are important data.
  • Line 130: “the % nominal values were between 85-115%, and the % RSD values were 130 less than 15% for all the stability samples.” The reviewer is afraid that this is not correct.
  • Line 175: “180.0 μg/mL” should be “180 μg/mL” to be consistent with “30, 60, …”
  • Line 179: “ACVR (IS) was” should be “ACVR (IR) were”.

Reviewer 2 Report

The article sounds like a typical laboratory report. To me any scientific novelty is present. Choosing the organic solvent for liquid-liquid extraction seems to be the one new part.

It is not mentioned what is really new in comparison with similar papers from the literature, e.g.:

1, Bulduk İ. HPLC-UV method for quantification of favipiravir in pharmaceutical formulations. Acta Chromatographica

  1. 冯光玲丁文娟邓玉晓赵仁永龚艳艳段崇刚孙晋瑞. A kind of Favipiravir has the HPLC assay method of related sub-287 stance. CN104914185B, 2016. 288

  1. 冯光玲丁文娟邓玉晓赵仁永龚艳艳段崇刚孙晋瑞. HPLC method for measuring related substances in Favipiravir. 289 CN104914185A, 2016.

The COVID-19 context is not explained at all.

Plasma of patients was used, but no Bioethics Commitee Agreement is included.

What is more, some self-citations are unclear, e.g.:

"Furthermore, the method was validated based on US-FDA regulations for Bioan-68 alytical Method Validation [29–37]."

31. Dayyih WA, Ani I al, Al-Shdefat RI, Zakareia Z, Hamid SA, Shakya AK. Development and validation of a stability-327 indicating HPLC method for empagliflozin and linagliptin in tablet dosage form. Asian Journal of Chemistry, 2021; 33(2): 328 484–8. 329

32. Abdel-Halim H, Dayyih WA. Pomegranate Juice-Drug Interactions: Pharmacokinetic Parameters Studied Using Dif-330 ferent Liquid Chromatography Techniques. Sapporo Medical Journal, 2020; 54(09). 331

33. Habash IW, Al-Shdefat RI, Hailat MM, Dayyih WA. A stability indicating rp-hplc method development for simul-332 taneous estimation of alogliptin, pioglitazone, and metformin in pharmaceutical formulations. Acta Poloniae Pharmaceu-333 tica - Drug Research, 2020; 77(4): 549–62. 334

34. Wael Abu Dayyih,Ramadan Al-Shdefat , Zainab Zakarya MH and GF. Determination of Rosuvastatin and 335 Dabigatran Simultaneously in Human Plasma for Application to Pharmacokinetic Parameters in Healthy Jordanian 336 Subjects by LC-MS/MS- Method. Journal of Pharmaceutical Sciences and Research (JPSR), 2019; 11(8): 2934–41. 337

35. Alanbaki A, Alani I, Mallah E, Zakareia Z, Arafat T, Abu Dayyih W. The effect of Pomegranate and Licorice on 338 Pharmacokinetics of Theophylline in Rat Plasma. Fabad Journal of Pharmaceutical Sciences, 2019; 44(1): 9–15. 339

36. Abu Dayyih W, Hamad M. Determination of sitagliptin levels in rats serum by HPLC and its pharmacokinetic in-340 vestigation in existence of sucralose. Indonesian Journal of Pharmacy, 2018; 29(3): 117–26. 341

Reviewer 3 Report

This is an interesting submission regarding the development and validation of a method for the quantification of Favipiravir as COVID-19 Management in Spiked Human Plasma. The method is interesting, however there are specific drawbacks of the submission

-“Of” is missing from the title (of a method)

-Authors omit citing previous methods and therefore do not compare the current one with existing protocols. This is very critical

-Real samples are missing. This is also very critical